# From Molecular Recognition to the “Vehicles” of Evolutionary Complexity: An Informational Approach

**DOI:** 10.3390/ijms222111965

**Published:** 2021-11-04

**Authors:** Pedro C. Marijuán, Jorge Navarro

**Affiliations:** 1Bioinformation Group, Aragon Health Sciences Institute (IACS), 50009 Zaragoza, Spain; 2Department of Quantitative Methods for Business and Economy, University of Zaragoza, 50006 Zaragoza, Spain; jnavarrol@unizar.es

**Keywords:** molecular recognition, informational architectures, information flow, signaling systems, biological complexity, evolutionary theory

## Abstract

Countless informational proposals and models have explored the singular characteristics of biological systems: from the initial choice of information terms in the early days of molecular biology to the current bioinformatic avalanche in this “omic” era. However, this was conducted, most often, within partial, specialized scopes or just metaphorically. In this paper, we attempt a consistent informational discourse, initially based on the molecular recognition paradigm, which addresses the main stages of biological organization in a new way. It considers the interconnection between signaling systems and information flows, between informational architectures and biomolecular codes, between controlled cell cycles and multicellular complexity. It also addresses, in a new way, a central issue: how new evolutionary paths are opened by the cumulated action of multiple variation engines or mutational ‘vehicles’ evolved for the genomic exploration of DNA sequence space. Rather than discussing the possible replacement, extension, or maintenance of traditional neo-Darwinian tenets, a genuine informational approach to evolutionary phenomena is advocated, in which systemic variation in the informational architectures may induce differential survival (self-construction, self-maintenance, and reproduction) of biological agents within their open ended environment.

## 1. Introduction

Opting for information to re-discuss evolutionary theory is not new. In the decades that followed the discovery of the DNA structure and the genetic code, information and communication terms were the metaphors of choice and Shannon’s was the exciting new theory to apply. Actually, “everyone was talking about information” in that time: Schrödinger, Shannon, Wiener, von Neumann, Turing, Watson, Crick, Dancoff, Quastler, Gamow, Monod, Jacob, among others (see quotations and references in [1]). Apart from copious vocabulary, perhaps the main achievement was the rigorous application of information theory to DNA, proteins, and cellular structures [2,3,4]. After the human genome sequencing, the connection with the informational apparently becomes tighter. In these recent decades, biology, directly, has been seen as an “information science” [5] for “the living is digital” [6], and bioinformatics, the “omic” disciplines, and systems biology, indeed, are part of the new “Big Science” [7] (see further references in [8]). This crescendo in the relationship between information and life has also occurred in physiology, biophysics, bioengineering, biomaths, ecology, computer science, etc., and has produced an enormous body of literature, particularly specialized in the bioinformatic and omic fields. To an important extent, the informational has also entered into the discussions around evolutionary theory. There seems to be a difference: while in the first epoch it was mainly supporting neo-Darwinian views in the molecular–biological arena (of relevance is Crick’s Central Dogma on the information flow), now, in the second epoch of the information metaphor, it has been handled by the challengers of neo-Darwinism as well. The remarkable discoveries and new knowledge about unconventional molecular mechanisms—from ‘facilitated variation’, to nonrandom mutations, ncRNAs’ role, retrovirus and virus integration, epigenetics, symbiosis, etc.—have compelled partisans of theoretical revision to also advance information and systemic arguments [9,10,11,12,13,14,15,16].

We might legitimately consider the living cell as an ‘informational entity’—as will be advanced in the next sections. Although we could find inspiration in other approaches based on Turing machines, operating systems, or other computer analogies, or even in the biosemiotic approach [17,18,19,20,21], we attempt to discuss the amazing informational capabilities of the living cell in a new way. The cell, we will discuss, is an open system, not only in the traditional thermodynamic sense, but also in terms of information. Therefore, we will complement the description of the *energy flow* in biological systems [22], essential in the thermodynamic view of the cell [23,24,25], by providing a parallel account of the cellular *information flow*. Departing from the water milieu and the multitude of molecular recognition phenomena, we will describe the basic informational architectures, the channeling of energy/information flows, and the integration of these signaling channels with gene expression and the whole self-construction within a tightly controlled cell cycle.

The cell cycle not only represents the reproduction characteristic, for it also realizes the projective relationship between genotype and phenotype. The reproduction cycle becomes, thus, an overall crossroads for all the fundamental properties of life and for its informational characterization—it is at the very basis of evolution and of biological complexity. The variety of systemic happenstances that could impinge on reproduction (plus self-construction and self-maintenance) in an open-ended environment will connect us with the fundamental discussions on evolutionary theory. Commenting on the recent discussions on the maintenance, renewal, or replacement of the Darwinian (neo-Darwinian) paradigm, we will draw a parallel between the mobility of genomes in sequence space and human mobility in physical space. We will consider how the different variation ‘vehicles’ might be lawfully interrelated, and how a plurality of selective and cooperative events might result in the differential survival of biological agents.

This will be our informational proposal. Our aim is not to solve sophisticated evolutionary problems, but to advance a new way of thinking, a general vision capable of reconciling artificially distant positions and narrowly focused approaches.

## 2. From Molecular Recognition to Informational Architectures

An elementary reverse engineering analysis of the simplest cells should give us essential cues. We will take *E. coli K-12* as our reference system.

When we ‘open’ the cell, what we foremost find is water, a significant amount of water. It constitutes 70% of its volume, around 20 billion molecules. As Szent Györgyi [26] put it (pp. 8–9): “water is in any case the most central substance of living nature. It is the cradle of life, the mother of life. It is our *mother* and *matrix*”. The form and function of all the other molecular species present in the cell bear the imprint of water: from DNA structure to RNA shapes, protein folding, enzyme function, membrane configuration, signaling molecules, metabolites, etc. [27]. Additionally, water appears as the computational *system bus* that transports most of the information flows via its intrinsic Brownian motion.

Further inside the cell, what we find is a gargantuan collection of biomolecules of an astonishing diversity—there is no possible comparison with any inanimate system. Approximately 2.5 million enzymes and proteins of 2000 different classes; a DNA circular polymer with more than 4.5 million base pairs and more than 4400 distinguishable genes (plus a variable number of plasmids); 300,000 RNAs of around 2000 classes; a collection of around 20,000 ribosomes; membrane phospholipids with around 20 million molecules of a few classes (4 plus 4); and finally a retinue of ions above the 100 million of 10 to 20 classes, as well as other organic molecules numbering around 3.5 million and of 800 classes or so [28] (figures rounded from [29]).

An incessant Brownian motion agitates the ensemble. Water is in relative sort supply regarding the appropriate *solvation* of all the organic molecules and its concentration has to be carefully controlled by aquaporins and mechano-sensitive channels in order to overcome the consequences of the Donnan effect (fatal osmotic shock). In this watery scenario, a myriad of molecular encounters take place at amazing velocities (cm/s to m/s within 1 μm diameter), exploring, almost instantaneously, hundreds or thousands of possible matching possibilities. They may appear random, but, indirectly, they are highly organized and controlled. First, as said, no “insulating wires” are needed; the watery diffusion *system bus* is one of the most remarkable processing resources of the living cell—a wired cell would be unthinkable of at the molecular scale. Secondly, the *specificity* of molecular recognition phenomena is the key to organize all that diversity of components into complex functions and transformative processes. It is at this point that our informational analysis starts.

### The Emergence of Informational Architectures out from Molecular Recognition Modes

The problem to start categorizing the myriad of biomolecular recognition events taking place within the cell is their sheer heterogeneity. They may involve an amazing variety and combinatorics of almost any type of chemical bond (and particularly of Coulombian motifs), which together provide the necessary specificity and affinity to the intermolecular matching encounters: covalent bonds, ionic Coulombian forces, hydrogen bonds, hydrophobic/hydrophilic forces, dipole forces, van der Waals forces, etc. Whatsoever molecular recognition event we may consider is but a chemical reaction, involving “the making and breaking of bonds”. Dozens or even hundreds of weak bonds may participate, for instance, in the recognition and aggregation of a protein–protein specific complex.

A few generalities may help in the categorization we need. First, those stablished in [30] about “biological homing”, contemplated from a Coulombian “lock-and-key” point of view, and that focus on the combination of motifs and the ensuing quantum interference effects that may economize recognition rates and accelerate reaction flows. Furthermore, those established in [31] about the changes in thermodynamic entropy and entropy of mixing derived from molecular recognition and molecular similarity, and also introducing the symmetry characteristics of *identity* and *complementarity* as main molecular recognition categories. In Table 1, we synthesize these two approaches and introduce *supplementarity* as a further category of molecular recognition [32]. Additionally, we considered those established in [33] about the design of recognition motifs for artificial nanosystems, distinguishing between recognition by *color* (e.g., sequences consisting in a few different motifs) and recognition by *shape* (lock-and-key motif arrangements) The pros and cons and the limitations of each procedure are analyzed and, importantly, the best outcomes derive from the integration of both shape and sequence procedures—as in some RNA types that do match by sequences and by specific loops. Finally, we considered the categories established in [34] regarding biological *informational architectures*, which are approached by considering their topology and dynamics. Although the approach is mainly focused on enzyme networks, the flows of information between different network architectures appear as an essential source of order for the working of the entire cell system (in particular for the advancement of its life cycle).

The above three ordering categories—identity, complementarily, and supplementarity—respectively mean: recognition by sharing *identical* molecular properties (e.g., self-organization of phospholipids in membranes, cytoskeleton structural networks, and vesicles and containment systems); recognition between the molecular partners by having *complementary* properties (e.g., molecular moieties, or the nucleic acids’ correspondence in the double helix); and finally recognition through the capability to envelop or *supplement* any molecular shape by building a complex molecular scaffold, usually of weak bonds, around the target (e.g., enzyme active sites and protein complexes). In the supplementarity case, the partial surfaces involved may be inherently sloppy in their specificity, having a variable affinity and even allowing competition with other similar substrates.

From an organizational point of view, these three categories abide well with the global distribution of functions within the cell and may be taken as the main classes of informational architectures (see Table 1). The different information flows between them and the “codes” established for their joint functions will be commented on later. For clarity, we will refer to them as *structural, sequential,* and *processing architectures.*

## 3. The Information Flow

As an open system regarding energy flows and information flows, the living cell presents us with the paradox that we can hardly distinguish between both types of flow, at least from a conventional thermodynamic point of view. We also ignore what classes or branches should be distinguished within information flows and how they interrelate with the energy flows (and matter flows) that are necessary for self-production.

Regarding the first aspect, the living cell organizes its processes of energy flows and information flows by following quite different strategies. As highlighted by Gerhardt [36], energy inputs in a metabolic pathway are processed through a series of enzymatic steps, being systematically depleted of their free energy until they finally emerge as an end product. However, in a signal transduction pathway, the input is not transformed at all; only an impulse is relayed by way of switch-like intermediates whose most frequent target is transcription. At least in prokaryotes, it happens that the most common signaling inputs detected from the environment are the very nutrients that will feed the metabolic pathways. Metaphorically, it means that inspecting the environmental affordances takes place prior to ingesting them. Alternatively, in a more conventional way, the high-energy, highly valuable flows related with the materials needed for self-production will be anticipated, detected, and captured by means of the faster and lighter communication flows tended with the surroundings. As a general rule, every ingested metabolic item always has to pass through a previous signaling control.

Regarding the classes to distinguish, we have to consider another information flow, from DNA to RNA to proteins, as the Central Dogma describes. By contraposition to our “external” information flow, we may call it “internal”. This internal information flow, we will see, is always disturbed or activated by the multiple signaling pathways that intercept the external information flow. Complete congruence is achieved in functional terms between these two information flows, the external and the internal. Actually, we will find that there is a partial physical mixing of both flows in the intracellular ”common pool” of substrates, products, and effectors. At the same time, the energy flow is materialized via the accesses opened by the signaling pathways and becomes incorporated into the intracellular common pool that supports self-construction and self-maintenance.

We have represented in Figure 1 the above general concepts: the external information flow, the internal information flow, and the energy flow. The red arrows of Figure 1 represent the effects of the 1CSs, 2CSs, and 3CSs signaling pathways to be discussed below. We may appreciate the overall congruence with the internal information flow (grey arrow) and with the energy flow inputs and outputs (blue arrows). Let us emphasize that the latter flow can take its proper form only after being detected and introjected by virtue of the signaling pathways below (concretely, a number of 2CSs and the entire 1CSs classes external and hybrid); then, it fully connects with the whole metabolic network and the “productive” common pool.

### 3.1. Prokaryotic Signaling: 1-2-3 Component Systems (CSs)

Let us examine in more detail all those signaling pathways by means of which the external information flow is regularly created. It will be performed within the general framework of prokaryotic signaling systems. The new knowledge recently gained about them is particularly important in our informational approach, and we will have to devote some extension to their scrutiny. Our model system continues to be *E. coli K-12.*

The variety of tools involved in environmental sensing—in organizing the external information flow—was put under the common term “signaling system” in the early 1990s [37,38]. A number of molecular apparatuses were contributing to this: receptors, transcription factors, processing enzymes, kinases and phosphatases, channels, transporters, etc. In the 1980s, the signaling pathway known as *“two component system”* was discovered, which was present in *E. coli* as well as in many other bacteria [39,40]. This system is composed of just two molecules: a receptor in the membrane and a response regulator inside. These two component systems (2CSs) were thought of as the great prototype of prokaryotic signaling and, as such, they have been subject of countless computing models and bioengineering schemes. Later, the *“one component systems”* (1CSs) were discovered [41,42,43,44]. They were far more numerous, detected far more metabolic substances, and worked only internally, fulfilling the role of both transcription factors and receptors crafted onto a single molecule. Further, *“three component systems”* (3CSs) were proposed as a complexification of the previous 2CSs [45,46]. Interestingly, this type of more complex pathway, which, for instance, regulates the motility of *E. coli*, had been previously studied as a molecular paradigm of neuronal processing properties [47].

Following Galperin [41], the predominance of one or the other class of signaling apparatuses distinguishes prokaryotic organisms concerned primarily with sensing environmental parameters (“extroverts”), well endowed with 2CSs and 3CSs, from other prokaryotic organisms more closely attuned to monitoring the internal milieu (“introverts”), highly endowed with 1CSs. Authors who have launched the field of “bacterial intelligence” [48] refer to the highly diverse external vs. internal sensitivity of these cells as cases of exceptionally high (or very low) bacterial “Intelligence Quotient”. For instance, *E. coli* counts with 30 different 2CSs, while other bacteria, such as *Streptomyces coelicolor*, counts with 53 2CSs, *Synechocystis* with 38, and *Pseudomonas aeruginosa* has at least 64 2CSs; conversely, *Chlamydia* counts with only one 2CS [49]. Bacteria having to survive both in free environments and in different parts of vertebrate guts may need faster and cleverer signaling strategies, as well as ampler metabolic flexibility. That is precisely the case of our model system, *E. coli.*
Table 2 shows the signaling figures corresponding to *E. coli K-12*.

The two 3CS are devoted to biofilm interactions and toxin/anti-toxin processes, while the thirty 2CSs pathways are devoted to metabolism, ion/metal, stress response, communication with conspecifics, and other unknown mechanisms (seemingly related to drug resistance). Together, they enable the bacteria to sense, respond, and adapt to a variety of circumstances: nutrient change, osmolarity variation, exchange of quorum signals, antibiotic resistance, temperature, pH variation, etc.

The ninety-two 1CSs classes, clearly more numerous, are categorized according to the provenance of the substances they detect: internal, hybrid, and external. See Table 3.

### 3.2. The Link between Signaling and Transcription

We can see in Figure 2 a good portion of the *E. coli* transcriptional system (almost half of the genes) that precisely responds to the above CSs [53,54,55]. We see represented in different colors (as described in detail in the legend of Figure 2) the distinct sensing realms that impinge upon gene transcription: the external sensed by 2CSs, in addition to the external transported, the hybrid external–internal, the purely internal, the global states, and the DNA sensing, most of them covered by 1CSs (following the order from left to right in the upper part of the Figure 2). We can also appreciate in Figure 2 that most transcription factors (TFs) play inhibitory roles (in red color), repressing, with high specificity, scores of genes already activated by sigma factors; their transcription will be allowed only when specific molecular signals activate those TFs (usually via 1CSs). There is a very important rationale about the predominance of inhibition. In fact, only 30% or 40% of the *E. coli*’s genome can be actively transcribed at any given time. There are several limits: the solvation capability of intracellular water, the transcriptional and translational functional capacities, and the tight energy budget of the cell. Given that less than 1/3 of the big genome can be expressed concurrently, a careful balance of inhibition/activation has to be imposed upon the transcriptional regime, with predominance of inhibition.

The beauty of this Figure 2 is that we can see, in its entirety, the signaling system of *E. coli K-12* coupled to its transcriptional targets. This is the most detailed representation we could find about the ”external” information flow intercepted by a prokaryotic cell and its retinue of transcriptional apparatuses. Figure 2 also represents the other information flow, from DNA to RNA to proteins, along the Central Dogma mandate. A “productive” common pool, in which metabolic and signaling contents are mixed after their respective, differentiated processing, may also be inferred. This productive common pool or “self-production flow” is a crucial ingredient for cellular self-construction and self-maintenance. It is crucial not only for the viability of the cell in a changing environment, but also for the information-based strategies of complexity growth.

The irretrievable final mixing of the external information flow with the internal information flow, and also with the energy flow, has fundamental evolutionary consequences: for inter-cellular problem solving, for the emergence of genuine multicellularity, and for the evolution of biological complexity. The external information flow will be able to change the entire self-production process of the living cell, including the DNA and RNA structures and processes. The opposite will occur as well, with the cell being able to emit ad hoc information flows congruent with its phase state and with the potential advancement of its own life cycle.

## 4. Integrated Functioning: The Cell Cycle and the Emergence of Meaning

All the systems and flows previously described contribute to obtain an adaptive life cycle that culminates in reproduction, the essential capability of life that permits the evolutionary process. While the life cycle is advanced, there appears another fundamental aspect of information: the generation of *meaning*. It is a highly important aspect of our informational approach in particular, and of information science in general. How can meaning be upended to Shannon theory, for instance? Current biosemiotic responses, we argue, are not satisfactory enough. In this paper, we focus on the continuous relationship with the environment, contemplated under the prism of what the signal “invisibly” conveys—its intrinsic meaning. We essentially argue that the meaning of a signal is fabricated by the receiver, and that the life cycle construct becomes the fundamental reference for its elaboration and for a cluster of associated concepts, such as value, relevance, and intelligence.

In our approach to meaning, all the descriptions of the previous section become essential. It is in the context of the advancing cell cycle, involving signaling and its subsequent transcription/translation processes, that meaning emerges. If we want to ascertain the effect that a given signal produces, say a particular portion of the external information flow, we must count the new molecular presences and absences derived from the gene-expression consequences of that signaling event. Thus, the meaning of that particular signal has to be established through what is called “molecular mining”. The living cell may systematically respond to whatever signal or signal coalitions of the environment (signaling “affordances”) and produce the meaning they imply, by letting the signals themselves circulate throughout the signaling system pathways and meddle with the ongoing self-production flow. Meaning has to be found in the (signal) induced changes: in transcriptional connectivity and in the activated enzyme–protein populations, plus associate metabolites and substrates, and above all in the subsequent protein synthesis. The *relevance* and *value* of impinging signals can subsequently be gauged within a crescendo of occurrences: metabolic buffering, second messengers’ alteration, transcriptional rewiring, and advancement through the cell cycle “checkpoints”. Completion of the cell cycle becomes the final and fundamental instance of successful reference.

Protein synthesis, which is the central activity of life, is at the same time the central fabric of meaning for the received signals in every tissue. Even in the most remote and highly evolved sites of organisms, such as excitatory synaptic spines, there continues to be local protein synthesis facilities [56,57]—the fundamental ‘material memory’ of life, the self-construction core.

The problem with the current semiotic and biosemiotic approaches to biological information is that, by convention, they have focused on DNA transcription and mRNA translation as the loci where meaning is established via the genetic code, from DNA sequences to proteins. For instance, the authors of [21] (p. 236) write: “The heart of all biological sign systems is the central dogma, describing how the DNA code (its signs mediated and processed as mRNA) becomes ‘translated’ by the tRNA/ribosome interpreting complex to form polypeptides”. It is unfortunate that these semiotic approaches do not focus on the direct relationship with the environment via signaling systems, in the many specific signals contained within the whole information flow that have to be interpreted and answered. As we argue in this paper, we should advance from the received signals to the realization of their meaning via the synthesis of ad hoc proteins and the whole changes in the self-production flow. Systematically, the realization of meaning adapts the living to its environment. Biosemioticians should be able to appreciate the beauty and relevance of the new knowledge gained about prokaryotic signaling systems, in which the link among the signaling transcriptional–translational processes becomes evident, in order to refine their highly partial approach to meaning.

The outcome of the cycle is not just another self-organization phenomenon: it is an informational life cycle with amazing properties of adaptability and evolvability within its open-ended environment. It mobilizes scores of signaling pathways, many metabolic resources, a multitude of inner self-construction processes, and, above all, it instantiates the capability of meaningful communication with other living agents.

### The Development of Further Life Cycle Complexity: Multicellularity

The amazing complexity later evolved by multicellular organisms was based in three main revolutionary developments. This organizational ‘new order’ involved optimizations of the energy flow (symbiogenesis), of the information flow (signaling explosion), and of the cell cycle (functional modularization). With these three achievements, the ontogenetic development could advance along unseen evolutionary complexification paths.

Indeed symbiogenesis implied an optimization of the energy flow. As Lynn Margulis [58,59] forcefully argued in her Theory of Endosymbiosis, when the eukaryotic cell became itself a symbiotic contraption or ‘composite’ of other cellular systems around 1200 Mys ago, the main change was energetic. Derived from the symbiotic capture of mitochondria, the efficiency gain was staggering: an average protozoan has nearly 5000 times more metabolic power than a single bacterium, and can support a genome several thousand times larger with more than two orders of magnitude in the energy devoted to expression and translation of each gene [60].

The processing of the external information flow was also optimized by the new composite cells. Their signaling expansion was basically due to processes of protein-domain recombination that allowed old prokaryotic resources and new eukaryotic tools to be put together within longer, mixed pathways that liberally cross-talked with each other. Osmotic tools (i.e., ion channels) were cobbled together with the detection of solutes by protein receptors and with hierarchic chains of protein kinases, as well as with the recycling of proteins in endosomes, connected with ubiquitination and degradation systems. Overall, just four main classes of functional resources were used in the expansion of eukaryotic signaling systems, four “roots” that supported the fast branching of all the new complexity [61,62]: signaling pathways devoted to detection of solutes, osmotic apparatuses devoted to solvent detection, hierarchies of protein kinases controlling the cell-cycle, and new cytoskeleton and endocytic matrix system effectors.

A new kind of modular life cycle, optimized for multicellularity, was produced. As a result of the signaling expansion, the main cycle phases were now amenable to controlled dissociation or modularization, making a new context of functional and tissular organization possible. Most cellular functions were changed from a temporal context to a spatial context, tightly controlled by specific signals; while some functions were delayed or directly suppressed, others became augmented and specialized [63]. The decoupling of cell division from cell reproduction, organizing successive levels of “potency” along the developmental process, was contingent on signals received from other cells, whereas on single cells these processes had no such dependence [36]. Thus, the capability to keep cells in a quiescent state by way of signaling instances, facultatively and reversibly, is what made the advent of true multicellularity possible [64,65].

In eukaryotes, the populational control of the different phases of the cell cycle (G1, S, G2, M) appears to be based on a cloud of internal and external signals, usually of opposed signs, carefully regulating the reproductive and specialization trajectories of cells and tissues—in general, growth factors versus apoptotic factors (see Figure 3). It is this balance between opposing factors, probably reflecting ancestral antiviral strategies [66], that propels cellular growth; eliminates transformed, senescent, or redundant cells; and keeps organs and tissues within their functional bounds. It is based on cross-road processes in which metabolic state, cell cycle state, and external signals are gauged together, converging in fundamental “checkpoints” where the fate of the whole cell is decided [67,68].

The most powerful set of protein kinases in the entire signaling system are directly associated with mitotic control: the MAPK cascade (MAPKKK, MAPKK, and MAPK). Depending on the cellular context, this cascade is divided into three branches: MAPK/ERK, SAPK/JNK, and p38/MAPK (see Figure 3) [61]. These kinase hierarchies may process an ample variety of inputs. Precisely, one of the functions attributed to MAPK cascades is to coherently insert a large variety of cross-talk signaling inputs from other pathways, endowing each one with the appropriate relevance and hierarchical order throughout the different signal amplification values of the successive kinase hierarchical levels.

From an informational perspective, multicellular organisms would have made an outstanding development. By means of their vast, hierarchic signaling systems, they can integrate, coherently, different information flows situated at different levels of organization. High level demands of the organism may be channeled as ad hoc signals to specialized populations of cells, becoming distinct components of their own external information flow. Once collected and transduced, the signals become irretrievably mixed with the productive common pool inside these cells, interfering with the ongoing self-production activities. Then, populations of lower-level agents are able to solve through their own adaptive mechanisms the corresponding portion of higher-level adaptive problems.

Multicellulars have evolutionarily mastered the methods of quasi-universal problem solving via differentiated tissues and specialized cells. In the way they are using force fields for top-down control of development [69], of bioelectric codes for early patterning [70], or in the sophisticate interplay of signaling gradients and gene expression [71], as well as in the use of gravitational force for structuring physiological fundamental axis [72], we may glimpse their quasi-universal organization capabilities, showing up in all kind of complex physiological functions. From the present informational perspective, should we not ponder about the awesome “engines” of genomic variability that have made possible all these informational capabilities?

## 5. An Informational Perspective on Evolutionary Theory

To paraphrase Dobszhanzy’s famous dictum, *nothing in biology would make sense except in the light of information*. Having substituted information by evolution, very few parties would nowadays deny that the former appears as the very ‘substance’ of the latter. Evolutionary theory itself may be the best place to substantiate that assertion, which indeed is the core of the present work. However, before this, the focus should be placed on another important informational/evolutionary theme, biological codes, which has been left in relative obscurity by most evolutionary thinkers and biosemiotic practitioners [73]. The evolution of codes and their most probable makers would return some of the fundamental information tenets in this paper.

### 5.1. On Codes and Code-Makers

In addition to the origins of the genetic code, still only partially solved [74], one of the most puzzling evolutionary developments is the fantastic variety of new codes achieved in eukaryotic evolution: from DNA genetic code to splicing codes, histone codes, sugar codes, compartment and transportation codes, cytoskeleton codes, tubulin code, nuclear signaling code, nuclear export code, ubiquitin and conjugating enzymes codes, epigenetic codes, adhesion codes, and many others [73,75,76,77]. According to the former author, prokaryotes that were unable to develop new codes precluded themselves from participating in the development of further organization complexity. The creation of new codes was necessary for the emergence of eukaryotes, it is clear, but what kind of events made them possible? Why in such a wide variety?

In the discussion about codes, we have to consider the general organization of eukaryotic complexity. Codes, in their most general acceptation, and in our own information views as well, mean the systematic correspondence between elements of two different worlds—in our case, distinct informational architectures—in order to organize complex functions with specificity and efficiency.

In the cellular milieu, there could be instances of codes between sequences matching to shapes, shapes matching to sequences; or shapes matching to shapes, and sequences matching to sequences; or there could be shapes within sequences, and sequences within shapes (enzymes and proteins have their own sequence of amino acids). Almost every complex cellular function, including ‘big codes’ such as the genetic code itself or the splicing code, could be decomposed on an assemblage of functional middle-level codes and low-level codes; all of them based on specific molecular-recognition events, quite often involving combinatorics of ‘colors’ (sequences, Section 2) and ‘shapes’ in between the architectural partners involved in each matching.

Thus, as in computer systems, we may speak about micro-codes, middle-level codes, and macro-codes. The assemblage of micro-codes and middle level codes that are needed to perform new complex functions after the command of a macro-code—in development, in physiology, or in neuronal processing—would be very difficult to achieve. In general, that will be possible only after the genomic incorporation of an external connoisseur or ‘expert’ bringing together of many of those matching events. The viral provenance of those external experts becomes highly probable. Along their convoluted circuits within cells, viruses have to recognize and interact with a number of protein factors and RNAs. For instance, in human cells, the HIV virus ‘knows of’ and specifically interacts with 453 human proteins [78]; the influenza virus interacts with 295 proteins [79]; the Ebola virus with 194 proteins [80]; and the new COVID-19 virus is said to interact with 332 proteins [81]. Additionally, these viruses and many others may hijack ncRNAs, lncRNAs, and miRNAs of very different classes in order to rewire cellular metabolism and to promote their own replication [82]. Let us emphasize the coding potential of many of those RNAs, as they may have not only ‘color’ or sequence, but also highly characteristic and diversified ‘shape’ in their loops, a factor that becomes crucial for their participation in complex matching populational processes.

If increasing the repertoire of cellular functions needed a loan about viable itineraries among architectures, there have been quite many possible ‘experts’ out there and in here. We must take into account that one the most important genome modifications of eukaryotes has come from the systematic activity of inner components of viral provenance: mobile elements, transposons, retrotransposons, repetitive elements, among others, which combined would represent more than 2/3 of the human genome [83]. Seemingly, our species has counted with around 4 million mobile insertion events [10]. Ancestral viral proteins can be found in human placental adhesion and development [84,85], in splicing machinery and in nuclear pores [11,66], in master gene regulators [86], and all across the mammalian and human proteomes [87].

Thereafter, it is almost inevitable to speculate about the participation of the whole “virome” in the evolutionary events leading to eukaryotic complexity and multicellular life [11,66,88,89,90,91] That means, among other things, the unexpected addition of a new, forgotten realm of life to Margulis’s endosymbiotic theory [57,58]. Perhaps more properly, it means plainly incorporating viruses’ essential evolutionary role within the present discussions around the renewal or replacement of evolutionary theory [91]. The “deep evolution” approach based on protein superfamilies analysis by hidden Markov models [92] would point to two independent lines of descent for eukaryotes and akaryotes (bacteria and archaea) out from the most recent universal common ancestor. Would eukaryotes constitute the more virus ‘friendly’ or labile line capable of fully integrating the viral code-making resources into the later complexity explosion? Ex virus omnia, according to [84].

The “Weismann barrier” was considered as isolating the germ cells’ genome from the phenotype vagaries. However, evolutionarily, we see it has been close to an ‘admission portal’—see, for instance, the penetration of ‘exosomes’ [12]. In general, the stages of germ cell, meiosis, gamete, fertilized zygote, gastrulation, and early development are highly susceptible to accommodate a variety of external/endogenous influences. Thus, the obligate return of multicellulars to the unicellular form becomes far more than intriguing [93], as it is at this stage when most of the above ‘travel companions’ are incorporated into eukaryotic vehicles, and when the strongest epigenetic effects occur. The phenotypic experiences of the parents in their separate exploration of the environment are sorted, modified by meiotic cross-over, and entered into the histone and DNA methylation codes. The mechanisms are still unclear, but the experimental evidence is there, in numerous species as well as in our own. The gamete as well as the zygote and the developing embryo are instructed about fundamental aspects of the panorama waiting out there—food scarcity, parental care, social environment, among others—and become subsequently preadapted through a metabolic–epigenetic-behavioral axis [94]. That almost all multicellulars reproduce by resorting to sexual gametes obtained after meiotic division has been a formidable engine both for evolutionary complexification and for consistently maintaining the stability of species [93].

As Torday [95] writes: “It is as if the unicellular state delegates its progeny to interact with the environment as agents, collecting data to inform the recapitulating unicell of ecological changes that are occurring. Through the acquisition and filtering of epigenetic marks via meiosis, fertilization, and embryogenesis, even on into adulthood, where the endocrine system dictates the length and depth of the stages of the life cycle, now known to be under epigenetic control, the unicell remains in effective synchrony with environmental changes”.

### 5.2. On Evolutionary Theory

In our informational vision, a sober analysis of the conventional evolutionary tenets would show troubles and absences. We have just seen two ample blocks of variability not very congruent with the classical Darwinian (or neo-Darwinian) formulation of evolution by random mutation and natural selection. The attempts to incorporate the known new facts as mere ‘add on’ to the conventional processes of evolutionary change have fallen short of the mark [96]. Nevertheless, their explanatory allure remains. The simplicity of neo-Darwinian views and their (apparently) universal explanatory reach, from molecular competition to the origins of life, to everything biological or human, makes them a magnet for reductionist thought of all kinds: “if it exists, it has been selected”.

Following [15], we will articulate our criticism by distinguishing between two main aspects: the generation of variability and the pruning or selection of that variability. Actually, although we seem to maintain two realms similar to the basic Darwinian views (variation and selection), we substantially change their contents, now formulated, we think, with more cogency regarding the sheer diversity of known facts.

Firstly, regarding generation of variability, we would be forced to an almost impossible compilation, putting together so many heterogeneous categories: mutations (neutral, significant, short-scale, and large-scale); sex and populations (allele distribution, recombination, and genetic drift); gene modification (horizontal transmission, frameshift change, intron/exon swapping, domain recombination, and duplications); chromosome rearrangements (translocations, inversions, deletions, crossover, and duplications); mobile, transposable, retrotransposable, and viral elements (with additional effects via ncRNAs, siRNAs, miRNAs, piwiRNA, and gene silencing); developmental (biased gene expression, repetitive DNA, ncRNAs, enhancers and regulators, and neoteny); epigenetic (histone code, DNA methylation, hormonal imprinting, and stress metabolic–behavioral axis); whole genome duplications; symbiosis; as well as a variety of other behavioral and environmental effects impinging on organisms.

At this point, we may consider a new term taken from human mobility studies: *containers* [97]. In the way human mobility is studied, one has to distinguish in which ‘container’ the movement is produced, and this means the corresponding standard range of displacements (distances) and frequencies (probabilities). So, in displacements by walking, cars, buses, metros, trains, planes, etc., each mode is producing an average displacement at an average frequency (or probability) for the average individual. They are not exclusive; for in a particular trip we may walk, take a taxi, a plane, and again a bus and some further walking. In general, and in aggregate, we will have separate ‘rectangles’ in the representation of displacement and frequency (probabilities) for each mobility mode, following a log-normal distribution. Now, the surprising effect, and the essential point about introducing the container term in our evolutionary discussion, is that when all containers are superimposed, the different rectangles disappear and a power law emerges regarding the aggregate human mobility. The introduction of new transportation modes would only contribute to change the slope of that power law. What this means is that the aggregate mobility of human population becomes self-similar, optimized regarding the classes of displacements and their frequencies. The smaller and slower displacements are far more common, but their aggregate mobility for the whole population is not too different from the medium range ones, which are less frequent, but cover longer distances; and the very fast modes are far less common, but more far-reaching in each displacement. This emerging commonality would recall the way in which energy is distributed equally among the different degrees of freedom of a molecule: linear velocities, rotations, oscillations, and vibrations. In fact, there is an equipartition between all of them (following what is called the “equipartition theorem”).

In a similar way, could we consider the aggregation of different “containers” regarding the evolutionary mobility of genomes in sequence space? Genomes have been moving quite a lot in sequence space, either walking at the minuscule pace of point mutations or at fantastic distances and speeds due to genome duplications or to symbiosis, with all kind of intermediate possibilities. In the neo-Darwinian exploration of adaptive landscapes, different kinds of exploratory motion around fitness peaks and valleys have definitely been considered: mutations, genetic drift, and sexual recombination [98]. Each mode conveys some specific displacement range in the adaptive landscape. Initially, thus, the comparison we are making between human mobility in physical space and genome mobility in sequence space does not look too dubious.

Further advancing the idea, we could group the multiple variability occurrences in just four containers: *mutational*, *genomic*, *developmental/epigenomic*, and *transgenomic*. In the evolutionary scenario, the less influential sources of variability, “mutational”, would be the most common mode; we have already mentioned the basic neo-Darwinian categories (mutations, genetic drift, and sex). The following container, “genomic,” could have more influential effects, but would be relatively less frequent; arguably, some parts could enter into the neo-Darwinian variation categories, but it would be very dubious for all the other gene, chromosome, and genome rearrangement categories, many of them due to transposon, retrotransposon, and viral agency. Furthermore, in the “developmental/epigenomic” container, we could group the most radical viruses’ effects regarding the development of novel tissues, functions, and codes, in addition to the multiple developmental (evo-devo) variability sources, as well as the epigenetic heredity phenomena. Finally, in the “transgenomic” (and also ‘metagenomic’) container, we could include the major symbiosis category and other not-so-radical effects due to external inter-actions beyond the genomic realm, such as behavior, niche construction, and different environmental effects—in general, the consequences of operating within an open-ended, interactive, and social environment.

Overall, in the genome evolutionary displacements, we should find that major evolutionary changes will be extremely influential, but extremely rare as well, so that in the long tempo of evolution the different sources of variability would have implied a similar cumulated ‘mobility’ in sequence space by each one. This is to say that, if the cumulated mobility of genomes in sequence space resembles human mobility, there would appear a smooth distribution of evolutionary change among the main variability containers—an equipartition. Every species would have followed its own trajectory at its own rhythm (the slope of its power law) propelled by the combined engines of variability it had accessed. However, not all containers should follow the same pattern of behavior, and what looks cogent for one of may be a blunder for others (e.g., forcing the “random” term for all mobile events). Additionally, counting with a variety of different components within each container, with say a range of distinct model “vehicles” of choice, could grant a smoother power law for their aggregate action. In the same way that complex societies have been developing more and more transportation modes for expanding individual motion in physical space, complex organisms would have developed more and more classes of evolutionary vehicles for their exploration of the increasingly vast sequence space.

The previous containers and their vehicles or engines of variability would be met by a series of eliminative counterparts, the different evolutionary sinks or selection events. Evidently, not just one class of “natural selection” occurs, but a vast plurality. This is to say that, in the same way that we can identify scores of different variation instances, there seems to be a similar range of diebacks. At least, we could distinguish environmental selection (e.g., due to physical parameters), ecosystem selection (e.g., altered habitat), niche selection (e.g., by invading species), predatory selection, parasitic/pathogenic selection, competitive selection, sexual selection, behavioral selection (Baldwin effect), social selection, group selection, keen selection, developmental selection, physiological purifying selection, stabilizing selection, among others. Could all of them be unified under the “natural selection” term? It is inconvenient for two reasons: that we hide the real causes intervening in the selective event (which is often clearly identifiable), and that selective instances may be purposive, do not belonging to some external nature, but to internal drives (e.g., sexual selection, social selection, and group selection). The efforts to prove that the ”natural” term applies to all of them following the Darwinian canon are futile—it becomes an ”artificial” label when applied to whatsoever selective causes. Additionally, we are ignoring the pervasive extension of *cooperative phenomena*. In quite a few of the selective instances mentioned, or in symbiotic relationships, cooperation of various kinds underlies the survival advantage gained by the organisms involved [99].

Natural selection is not a source of adaptive design either, actively pruning the randomness of mutational events and leaving a string of progressive functional achievements, as claimed. Necessarily, the core design itself—or a viable precursor—has to be produced in advance by several combined variation events, by natural experimentation, without excluding instances of channeled or *directed evolution*. Darwinism only circumscribes the demographic fate of novelties, which in a variety of “easy” instances may lead to speciation (e.g., first container and parts of the second). However, selection is not merely the result of a competition of alleles within populations. In its more general acceptation, it would mean *differential survival*, including the whole self-construction, self-maintenance, and reproduction effects, which may be due to a plurality of causes, often identifiable. The trouble with maintaining the conventional term is that too much ideological, unscientific reasoning has been collected under the rug of natural selection. It is not only the preaching of a few ultra-Darwinists, such as Richard Dawkins and Peter Atkins. For instance, the latter has penned an astonishing sentence [100]: “A great deal of the universe does not need any explanation. Elephants, for instance. Once molecules have learnt to compete and to create other molecules in their own image, elephants, and things resembling elephants, will in due course be found roaming around the countryside... Some of the things resembling elephants will be men”. Amen? Unfortunately, a large portion of the educated public, and indeed many practitioners within biological fields, still uncritically accept and follow the Darwinian dogma of *omnia ex nihilo* [15]. The overextension of the Darwinian paradigm of evolution by random mutation and natural selection outside the borders of its ‘natural containers’ has become another of the “Great Blunders of Science” [101], joining *entropy*, *information*, and *cognition as computation* as one of the most vexed conceptualizations of our time.

Although some easy parts of the origin of species problem have been solved via the Darwinian tenets, there remains too much evolutionary complexity to adumbrate, to explain, and to accommodate in a parsimonious formula. Alternatives to the beguiling simplicity of Darwinian and neo-Darwinian views are badly needed and should be actively looked for. Recently, a discussion has been taking place, for instance, between Neo-Darwinian defenders and those who consider the renewal of evolutionary theory just as a matter of paradigm extension [96], in addition to others who consider that a full replacement is needed [12]. Our informational proposal would join the latter views, though trying to change the reference framework. The alternative at the time being should not be limited to a relatively long list of thematic discrepancies and agreements; a new way of thinking is needed that is capable of a parsimonious synthesis. This should also include looking for a competitive short formula that is crafted tentatively to cover the two fundamental aspects of evolution: generation of variability and pruning by selective processes.

Variation, we have seen, might be grouped into four containers, each one having quite a few different vehicles inside. The variation resulting from their individual and aggregate action, including ‘forbidden’ feedbacks with the phenotype and with the environment, can be aptly categorized as *systemic*. This is probably the most inclusive term. In this way, we might talk about “systemic variation”. It has to be accompanied by the term “informational architectures”, for these architectures are the scenario in which the systemic variation finally becomes gauged and registered. On the other side, we may point out that there is a differential survival impact, in relation to the ongoing processes of self-construction, self-maintenance, and reproduction, which are always taking place in the background of an interactive and social environment.

So, we could synthesize the formula: “evolution proceeds by systemic variation in the informational architectures, which may bring forth the differential self-construction, self-maintenance, and reproduction of biological agents within their open ended, interactive environment”. Put in much shorter terms: evolution by systemic variation and differential survival.

## 6. Conclusions: Cogence of the Informational Approach

In this paper, we attempted a bottom up panoramic that started in molecular recognition and progressed towards informational architectures, information flows (external and intrinsic), and the coupling of signaling systems and gene expression, with everything converging on a central element, the life cycle. Our further excursion on eukaryotic complexity and evolutionary theory would deserve a more detailed discussion, but we think that the few milestones achieved in the paper could delimit a new thinking space.

Molecular recognition, surprisingly, is not a mainstream topic in itself, though it secondarily appears in countless biomolecular, bioinformatic, and biophysical experimental works. It was chosen as our starting point, conversely to many other approaches in the informational genre, mostly focused on Shannon theory, Bayesian theory, network approaches, and all those sophisticate bioinformatic algorithms BLAST style (technologically irreproachable). In what follows, we add emphasis on three points of significance.

First, the novelties on prokaryotic signaling should be emphasized. The new knowledge gained about prokaryotic signaling systems, the whole 1-2-3 Component Systems, is clearly showing the close relationship between energy flow and information flow. There is a direct, vast link from signaling to transcriptional processes and to translational processes, which has been traditionally ignored in the classical information flow of the Central Dogma. This link makes it patent that protein synthesis, which is the central activity of life, is at the same time the central fabric of meaning for received signals. The many specific signals contained within the whole external information flow have to be interpreted and answered, and the systematic realization of their meaning is at the same time what adapts the living to its environment. Biosemioticians should pay more attention to cellular signaling systems (as well as to what we have said about “codes”). Protein synthesis is a hallmark for all living activities. Without local protein synthesis, for instance, our postsynaptic spines would be ”flat”, unable to adapt their coupling strength to concordant excitations.

Second, we focused on the life cycle. It is the most amazing information design that any engineer could conceive. The living cell is a system that self-constructs out from environmental matter according to an inner blueprint that is separate from the constructive system itself (echoing von Neumann’s self-reproducing automata). With unencumbered repetition of the reproduction cycles, there is a tendency to excess, to fill in the ecological niche, but the emerging trophic interactions will put all participants “in their place”. Furthermore, systemic variations affecting the blueprint will appear, becoming phenotype changes and implying differential survival; thus evolution occurs. Along the evolutionary process, quite many variation ‘vehicles’ will be assembled in multicellular organisms for the adaptive exploration of the new complexity scenarios, implying both modifications of the DNA blueprint and interactive behavior in the coupled environment.

Finally, third, we emphasized the need of a new evolutionary synthesis, with an informational bend. Indeed, the cross-fertilization in between the biological and the informational realms seems to have occurred in a biased way, mostly from the former towards the latter (genetic algorithms, perceptrons, and neural networks). The enormous technological loan via bioinformatics and “omic” disciplines in recent times has been rather thrifty at conveying new scientific thought. Probably, the lack of an enlarged and consistent information science has been an insuperable obstacle to properly cross-fertilize with biological and evolutionary fields.

In the interim, whether we are close to a proper theoretical solution or not, putting to test the evolutionary proposal drafted in this paper about a power law smoothly interconnecting the whole evolutionary “containers” and “vehicles” might be a more than interesting exercise.

## Figures and Tables

**Figure 1 ijms-22-11965-f001:**
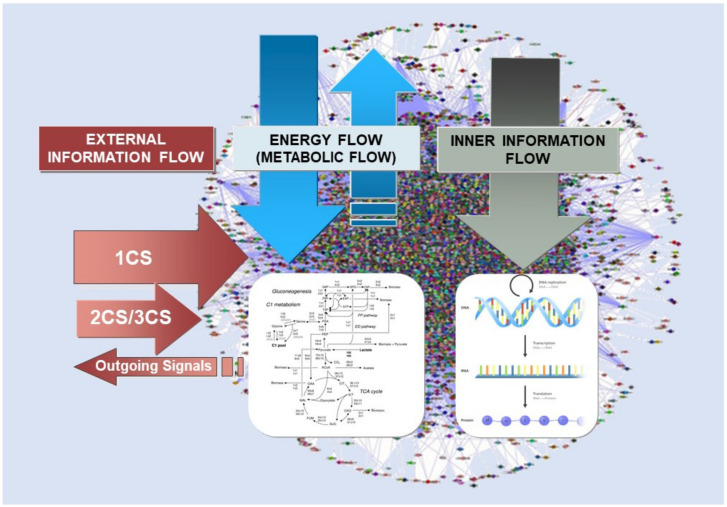
Energy flows and information flows in the living cell. They are shown respectively as blue arrows in the center part (energy flow), grey ones in the right part (inner information flow), and red arrows in the left part (external information flow).

**Figure 2 ijms-22-11965-f002:**
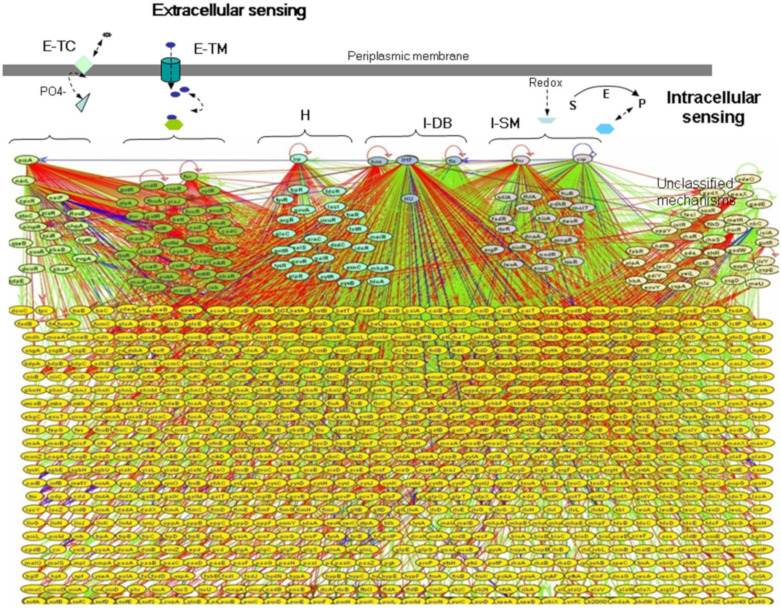
The *Escherichia coli* transcriptional regulatory network for sensing the extracellular and intracellular environment. In the upper part and from left to right, in green, are those transcription factors (TFs) corresponding to the *extracellular class of sensing*; in light green, are those TFs from *two-component systems* (E-TC) and, in dark green, are those TFs using exogenous metabolites transported into the cells by *transport systems* (E-TM). In light blue are those TFs corresponding to *hybrid system of sensing* (H), i.e., those TFs using metabolites synthesized inside the cell and incorporated from the milieu. In dark blue are those TFs for DNA-bending or chromatin architectural TFs; they do not sense metabolites directly. In pink are those TFs for sensing intracellular conditions or sensing the internal cellular redox-state. Finally, in light orange are those TFs without metabolites or unknown mechanisms to modulate their activities. Global TFs (ArcA, Lrp, Hns, IHF, FIS, FNR, and CRP) are at the top level. The connecting lines in green represent activation; in red repression; and in blue dual (activation and repression). In yellow (below), there appear numerous functional genes that do not code for TFs. Other abbreviations: S, substrate; E, enzyme; and P, product. Reproduced from RegulonDB [55], with permission.

**Figure 3 ijms-22-11965-f003:**
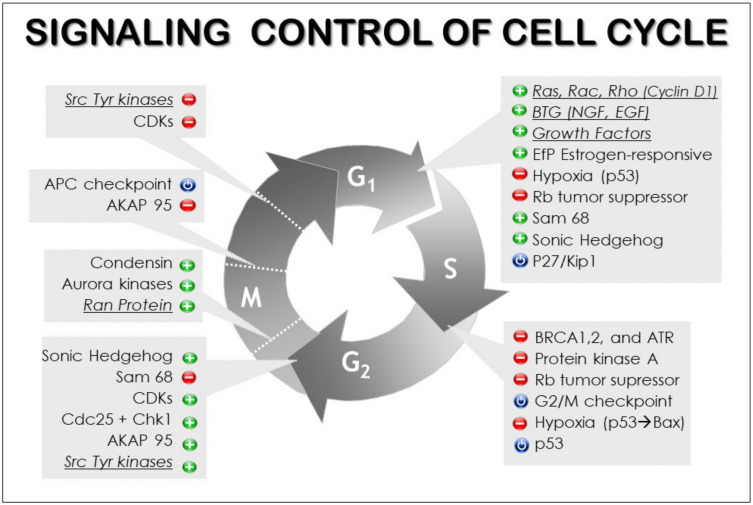
Cell cycle control. A tight signaling control is established on the different phases of the cell cycle (*G*1, gap; S, synthesis; *G*2, interphase gap; and M, mitosis) and on their respective transitions. The modular organization of the multicellular organism allows the space–time separation between cell cycle phases. The underlined pathways are connected with the MAPK cascade.

**Table 1 ijms-22-11965-t001:** Basic categories of molecular recognition in the living cell. Together, these three architectural classes integrate the “universal processing and self-constructing system” of the living cell, capable of exploiting an endless variety of boundary conditions at the molecular scale in the pursuit of the life cycle, including the capability of ‘meaningful’ communication with the environment. (Note: some items in the Table 1 correspond to eukaryotes.) Modified from [35].

Identity (Structural)	Complementarity (Sequential)	Supplementarity (Processing)
nucleotides/RNA	RNA/RNA pairing	enzymes/substrates
nucleotides/DNA	RNA/DNA pairing	enzymes/effectors
amino acids/protein chains	RNA/ribozymes	enzymes/cofactors
phospholipids/membranes	RNA/ribosomes	receptors/ligands
tubulins/microtubules	RNA/ribonucleoproteins	channels/ions
actins/microfilaments	DNA/DNA pairing	channels/ligands
clathryn/vesicles	DNA/polymerases	antibodies/antigens
carbohidrates/glycoproteins	DNA/promoters	proteins/chaperons
lipids/lipoproteins	DNA/histones	proteins/proteasomes
histones/nucleosomes	DNA/transcript. factors	proteins/converter enzymes
proteins/protein multimers	DNA/repressors	proteins/protein complexes

**Table 2 ijms-22-11965-t002:** Signaling summary of *E. Coli K-12*.

Signaling Component Systems	
**Three Component Systems (2)**	Complex communication processes
**Two Components Systems (30)**	Communication with the environment
**One Component Systems (92)**	For import external compounds (28) For import and processing of hybrid metabolites (33) For processing of internal metabolites (26) For detection of global states (4 DNA binding, 1 redox)
**Sigma Factors (7 + 14)**	Sigma factors (+ anti-sigma factors + anti-anti-sigma factors)

**Table 3 ijms-22-11965-t003:** Functional classes of One Component Systems (1CSs).

*External compounds 1CSs* (28), devoted to detect inside the cell external environmental compounds that may be important for cell survival: ions/metals, vitamins, cofactors, antioxidants, etc. For instance, transcription factors (TFs), such as ArsR, CueR, ExuR, FucR, BetI, and AgaR, constitute 1CSs respectively devoted to the detection inside the cell of environmental As/Cd, Cu, hexuronte, fucose, choline, and GalNAc/GalN. They have crossed the semipermeable membrane (directly or via transporters), but, after being detected, their import is augmented via the transcription and translation of specific transporters.
*Hybrid compounds 1CSs* (33), devoted to the detection of compounds that may be found either externally in the environment or internally within the metabolic pathways. Actually, the cell ‘ignores’ the inner/outer provenance of these compounds, so it transcribes and translates both the transporters to import them and the enzymes needed for their metabolic processing. For instance, TFs, such as Arac, AsnC, DsdC, GalS, GlcC, and GlpR, constitute 1CSs respectively devoted to the detection of arabinose, asparagine, serine, galactose, glycolate, and glycerol-3P. The legendary Lac operon discovered by Jacob and Monod is a paradigmatic case within this category [50].
*Internal compounds **1CSs* (26), devoted to sensing exclusively internal metabolites. In this case, the activated TFs transcribe the enzymatic genes needed to empower the respective pathways so as to maintain a consistent structure of metabolic flows. This level of control, based on the ad hoc presence of substrates, products, and effectors (activators and inhibitors), detects the absence of necessary processing enzymes and restores an adequate balance. For instance, TFs, such as BirA, CRP, FarR, IciA, LeuO, and MalT, constitute 1CSs respectively devoted to the detection of biotin-5P, cAMP, acyl CoA, ATP, ppGpp, and maltotriose.
*Other 1CSs and Sigma Factors* (around 40), this additional class would include the detection of global DNA states as well as the activity of sigma factors that oversee big transcriptional ‘moods’, and it could also include the sensory capabilities due to bacterial ion channels, which might be activated via mechano-sensing, voltage, or by associated receptors [51,52]. There is also the emission of signals to the environment, when the cell communicates with conspecifics or with other species within microbial ecosystems, or when it manipulates the signals of their hosts—the most important outgoing products or signals are exported via specialized secretases.

## Data Availability

Not applicable.

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
