# Peer review of "From Molecular Recognition to the “Vehicles” of Evolutionary Complexity: An Informational Approach"

_ijms, 2021, doi:10.3390/ijms222111965_

Round 1

Reviewer 1 Report

This work is a very complex and stimulating manuscript, where the authors propose an extended perspective about informational architectures, information flows, and the evolution of complexity. One of the most interesting points is the need for an evolutionary synthesis with an informational bend.

From my perspective, the major issues of the work are:

  1. The excessive use of analogies or metaphors. Mainly, the interpretation, discussion, and significant conclusions are made using the analogies as facts or an argument. In any case, the perspective is important and could open a significant discussion in the field, and the manuscript is a lineage of interesting work made by the authors and will be inspiring for the audience interested in the topic.
  2. A wide range of topics is covered in the text. A succinct version based on the information flow and the "vehicles" of evolutionary complexity would be more productive and clearer.
  3. I agree with the authors in not having an ultra-Neo-Darwinist version. However, a strong confusion about positive-negative Natural Selection is made in the 7.2 section, which must be clarified or deleted from the text. Furthermore, the conclusion about this point is very Darwinian (no Neo-Darwinist) view, at least in the "differential survival."

As mentioned above, the main problem is the extrapolation of the analogies as a fact. Therefore, it is essential to insist that Vocabulary, Syntax, Semantics, even molecular communication are metaphors but are only analogies. Nevertheless, the perspective, discussion, and future implications will be critical in the field. Thus, I suggest less assertive and more insinuate wording.

Author Response

This work is a very complex and stimulating manuscript, where the authors propose an extended perspective about informational architectures, information flows, and the evolution of complexity. One of the most interesting points is the need for an evolutionary synthesis with an informational bend.

Thanks for your review, and for the general positive opinion about our work.

From my perspective, the major issues of the work are:

  1. The excessive use of analogies or metaphors. Mainly, the interpretation, discussion, and significant conclusions are made using the analogies as facts or an argument. In any case, the perspective is important and could open a significant discussion in the field, and the manuscript is a lineage of interesting work made by the authors and will be inspiring for the audience interested in the topic.

In our informational approach most of the terms used are not intended as metaphors. For instance, the information flow, informational architectures, or cellular communication, or the elaboration of meaning. In the same way that we have an accepted “biophysics” dealing with the energy flow and all the related bio-physical phenomena, we really need a “bioinformation” dealing with the plethora of informational phenomena occurring in cells and organisms—still not developed. We accept that in the evolutionary discussion the terms “vehicles” and “containers”, taken from human mobility, may be seen as metaphors more or less cogent, but they are at least attractive to highlight the different kinds of genomic displacements in sequence-space.  

  1. A wide range of topics is covered in the text. A succinct version based on the information flow and the "vehicles" of evolutionary complexity would be more productive and clearer.

Thanks. We have deleted 4,000 words. And now there are only four sections. We have maintained “molecular recognition” as it is important for making sense of informational architectures, codes, and evolutionary role of viruses.

  1. I agree with the authors in not having an ultra-Neo-Darwinist version. However, a strong confusion about positive-negative Natural Selection is made in the 7.2 section, which must be clarified or deleted from the text. Furthermore, the conclusion about this point is very Darwinian (no Neo-Darwinist) view, at least in the "differential survival."

We basically follow R. Reid (2007) views on selection as a multicomponent, multidimensional process. If we analyse fitness (the changes of) after some genomic variability events, or after changes in the environment, we would find that some of the related factors are producing positive gains while others are implying negative values. But there is an overall result, with either a positive or a negative value. So, we do not think that the 7.2 text is distanced from that fundamental idea. The importance of “differential survival” (better captured in the extended formula; “evolution proceeds by systemic variation in the informational architectures, which may bring forth differential self-construction, self-maintenance, and reproduction of biological agents within their open ended, interactive environment.”) is that it dispenses of “natural selection” and emphasizes the multiple factors that may bring selection pressures.

As mentioned above, the main problem is the extrapolation of the analogies as a fact. Therefore, it is essential to insist that Vocabulary, Syntax, Semantics, even molecular communication are metaphors but are only analogies. Nevertheless, the perspective, discussion, and future implications will be critical in the field. Thus, I suggest less assertive and more insinuate wording.

We have not used those terms of Vocabulary, Syntax or Semantics. But, OK, we have softened some paragraphs (or just eliminated them), particularly in the Intro and in the Conclusions. We think it may now read more acceptably.

Reviewer 2 Report

The authors tried to describe a biological system in terms of information flow/structure. They tried to provide examples of how specific biological entities map to information structures and flows, not in a metaphorical way. It is an interesting attempt and makes some sense, but I  think it does not fit well with the scope of this journal. Perhaps it would be better to publish a paper in a more specific journal. Also, it is not clear whether this submitted paperis a review paper or an article. 

Issues
There are numerous correct trackings to make it difficult to read.
Fig2 and Fig3 are the same figures that they used in their previous paper. It would be better to change at some point, if this is an article.

Added: "If the category of this paper is an opinion, essay, or review paper, then I could understand the format and the way to write. But it doesn't fit my criteria to be considered as a scientific article. If you want to prove certain hypothesis, I think you need to also address the alternative hypothesis as well. The way to presenting here is just description of their opinion with some examples. It is really difficult for me to say the hypothesis is correct or not, but it is just the author's opinion. And the point of view that cell is information entity is covered before as the author described in abstract and introduction. This paper is try to incorporate more molecular mechanism. But, as it is just description, for me it is difficult to evaluate it with any scientific method. As a philosophical subject, I feel that it is beyond my ability to evaluate it. Therefore, I recommend sending this paper to other reviewers to receive an evaluation.
Minor issues I found: Title. 3.1. Modularity and Networking -> change 3.1 to 3.2
Title. 7. The evolutionary growth of complexity -> Same title as 6. The evolutionary growth of complexity
Protein synthesis is a hallmark for all the living activities. Without local protein synthesis, for instance, our postsynaptic spines would be 'flat', unable to adapt their coupling strength to concordant excitations. => Yes, it is true that protein synthesis is a key of living activities. But I think the example is here is very specific and narrow. "

Author Response

The authors tried to describe a biological system in terms of information flow/structure. They tried to provide examples of how specific biological entities map to information structures and flows, not in a metaphorical way. It is an interesting attempt and makes some sense, but I think it does not fit well with the scope of this journal. Perhaps it would be better to publish a paper in a more specific journal. Also, it is not clear whether this submitted paper is a review paper or an article. 

Thanks for your review work. Concerning this initial point, the special issue is about “From Nanomachine to Nanobrain, Information Processing at a Molecular Scale”, so inviting toward new views more open to informational aspects—isn’t it? Well, we recognize that mostly experimental journals may be reluctant to handle theoretical-interdiciplinary work of this kind. And the converse occurs as well, with theoretical journals often not being interested in works too closer (for them) to experimental matters. So we are glad about participating in this special issue. But in any case we will accept the final opinion of the Journal about that.

Issues
There are numerous correct trackings to make it difficult to read.
Fig2 and Fig3 are the same figures that they used in their previous paper. It would be better to change at some point, if this is an article.

Thanks. We have modified them.

Added: "If the category of this paper is an opinion, essay, or review paper, then I could understand the format and the way to write. But it doesn't fit my criteria to be considered as a scientific article. If you want to prove certain hypothesis, I think you need to also address the alternative hypothesis as well. The way to presenting here is just description of their opinion with some examples. It is really difficult for me to say the hypothesis is correct or not, but it is just the author's opinion. And the point of view that cell is information entity is covered before as the author described in abstract and introduction. This paper is try to incorporate more molecular mechanism. But, as it is just description, for me it is difficult to evaluate it with any scientific method. As a philosophical subject, I feel that it is beyond my ability to evaluate it. Therefore, I recommend sending this paper to other reviewers to receive an evaluation.

It is quite right that this is a work closer to “an opinion, essay, or review paper” than to a regular scientific work.  There is a point of some importance about this. About 60 or 70 years ago, biophysics was a rarity, something to be searched in the outskirts of the establishment (the brutal discovery by Watson & Crick) dramatically changed things. Well, in the same way that we have nowadays completely accepted “biophysics” dealing with the energy flow and all the multiple bio-physical and molecular phenomena, we imply herein that we really need a “bioinformation” discipline dealing with the plethora of informational phenomena occurring in cells and organisms—still far from being developed. To put another example, Quantum Computing is having a great companionship with quantum information science... the former more applied and experimental while the latter is more theoretical and fundamental. This style of discipline complementarity is needed regarding the boom of bioinformatic and biocomputing developments versus a more theoretical and fundamental bioinformation discipline—which, I insist, unfortunately still does not exist.

Minor issues I found: Title. 3.1. Modularity and Networking -> change 3.1 to 3.2
Title. 7. The evolutionary growth of complexity -> Same title as 6. The evolutionary growth of complexity

Thanks for these corrections. They have been made (and in some cases eliminated as now the manuscript is considerably shorter, in 4,000 words)

Protein synthesis is a hallmark for all the living activities. Without local protein synthesis, for instance, our postsynaptic spines would be 'flat', unable to adapt their coupling strength to concordant excitations. => Yes, it is true that protein synthesis is a key of living activities. But I think the example is here is very specific and narrow. "

OK, we have enlarged a tiny bit the point (also in section 3).

Round 2

Reviewer 2 Report

I think the the manuscript is significantly improved. There are minor things to be corrected.

8. Conclusion: cogence of the informational approach ==> 6. Conclusion